# QuAnt: Quantum Annealing with Learnt Couplings

**Marcel Seelbach Benkner**
Universität Siegen

**Maximilian Krahn**
MPI for Informatics, SIC
Aalto University

**Edith Tretschk**
MPI for Informatics, SIC

**Zorah Lähner & Michael Moeller**
Universität Siegen

**Vladislav Golyanik**
MPI for Informatics, SIC

## ABSTRACT

Modern quantum annealers can find high-quality solutions to combinatorial optimisation objectives given as quadratic unconstrained binary optimisation (QUBO) problems. Unfortunately, obtaining suitable QUBO forms in computer vision remains challenging and currently requires problem-specific analytical derivations. Moreover, such explicit formulations impose tangible constraints on solution encodings. In stark contrast to prior work, this paper proposes to learn QUBO forms from data through gradient backpropagation instead of deriving them. As a result, the solution encodings can be chosen flexibly and compactly. Furthermore, our methodology is general and virtually independent of the specifics of the target problem type. We demonstrate the advantages of learnt QUBOs on the diverse problem types of graph matching, 2D point cloud alignment and 3D rotation estimation. Our results are competitive with the previous quantum state of the art while requiring much fewer logical and physical qubits, enabling our method to scale to larger problems. The code and the new dataset are available at https://4dqv.mpi-inf.mpg.de/QuAnt/.

## 1 INTRODUCTION

Hybrid computer vision methods that can be executed partially on a quantum computer (QC) are an emerging research area (Boyda et al., 2017; Cavallaro et al., 2020; Seelbach Benkner et al., 2021; Yurtsever et al., 2022). Compared to classical methods, they promise to solve computationally demanding (*e.g.,* combinatorial) sub-problems faster, with improved scaling, and without relaxations that often lead to approximate solutions. Although quantum primacy has not yet been demonstrated in remotely practical usages of quantum computing, all existing quantum computer vision (QCV) methods fundamentally assume that it will be achieved in the future. Thus, solving these suitable algorithmic parts on a QC has the potential to reshape the field. However, reformulating them for execution on a QC is often non-trivial.

QCV continues building up momentum, fuelled by accessible experimental quantum annealers (QA) allowing to solve practical ($\mathcal{NP}$-hard) optimisation problems. Existing QCV methods using QAs rely on analytically deriving QUBOs (both QUBO matrices and solution encodings) for a specific problem type, which is challenging, especially since solutions need to be encoded as binary vectors (Li & Ghosh, 2020; Seelbach Benkner et al., 2020; 2021; Birdal et al., 2021). This often leads to larger encodings than necessary, severely impacting scalability. Alternatively, QUBO derivations with neural networks are conceivable but have not yet been scrutinised in the QA literature.

In stark contrast to the state of the art, this paper proposes, for the first time, to learn QUBO forms from data for any problem type using backpropagation (see Fig. 1). Our framework captures, in the weights of a neural network, the entire subset of QUBOs belonging to a problem type; a single forward pass yields the QUBO form for a given problem instance. It is thus a meta-learning approach in the context of hybrid (quantum-classical) neural network training, in which the superordinate network instantiates the parameters of the QUBO form. We find that sampling instantiated QUBOs can be a reasonable alternative to non-quantum neural baselines that regress the solution directly.

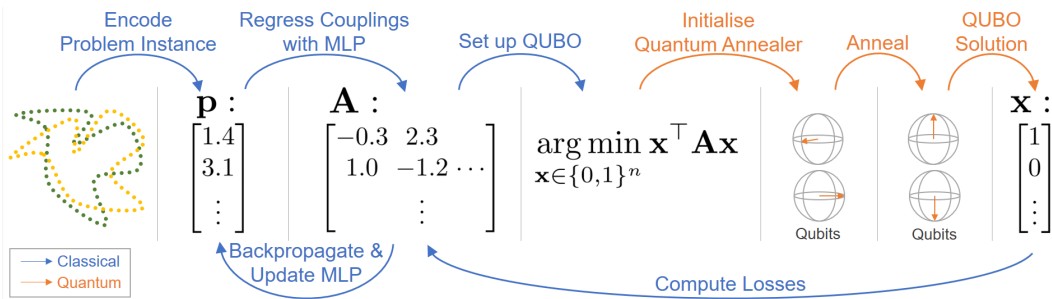

Figure 1: We propose **QuAnt for QUBO learning**, *i.e.,* a quantum-classical meta-learning algorithm that avoids analytical QUBO derivations by learning to regress QUBOs to solve problems of a given type. We first represent a problem instance as a vector $\mathbf{p}$ and then feed it into an MLP that regresses the entries of the QUBO matrix $\mathbf{A}$. We then initialise a quantum annealer with $\mathbf{A}$ and use quantum annealing to find a QUBO minimiser and extract it as the solution $\mathbf{x}^*$ to the problem instance. We define losses involving $\mathbf{x}^*$ that avoid backpropagation through the annealing and backpropagate gradients through the MLP to train it. We demonstrate the generalisability of QuAnt on graph matching, point set registration, and rotation estimation.

In particular, we show how a (combinatorial) quantum annealing solver can be integrated into a vanilla neural network as a custom layer and be used in the forward and backward passes, which may be useful in other contexts. To that end, we introduce a contrastive loss that circumvents the inherently discontinuous and non-differentiable nature of QUBO solvers. Our method is compatible with any QUBO solver at training and test time—we consider parallelised exhaustive search, simulated annealing, and quantum annealing. QUBO learning, *i.e.,* determining a function returning QUBO forms given a problem instance of some problem type as input, is a non-trivial and challenging task. In summary, this paper makes several technical contributions to enable QUBO learning:

1. QuAnt, *i.e.,* a new meta-learning approach to obtain QUBO forms executable on modern QAs for computer vision problems. While prior methods rely on analytical derivations, we learn QUBOs from data (Sec. 3.1).

2. A new training strategy for neural methods with backpropagation involving finding low-energy solutions to instantaneous (optimised) QUBO forms, independent of the solver (Secs. 3.2 and 3.3).

3. Application of the new framework to several problems with solutions encoded by permutations and discretised rigid transformations (Secs. 3.4 and 3.5).

We show that our methodology is a standardised way of obtaining QUBOs independent of the target problem type. This paper focuses on three problem types already tackled by QCV methods relying on analytical QUBO derivations: graph matching and point set alignment (with and without known prior point matches in the 3D and 2D cases, respectively). We emphasise that we do not claim to outperform existing specialised methods for these problem types or that QA is particularly well-suited for them. Rather, we show that this wide variety of problems can be tackled successfully and competitively by our general quantum approach already now, before quantum primacy. Thus, in the future, computer vision methods may readily benefit from the (widely expected) speed-up of QC through an *easy* and *flexible* re-formulation of algorithmic parts as QUBOs, thanks to our proposed method. We run our experiments on D-Wave Advantage5.1 (Dattani et al., 2019), an experimental realisation of AQC with remote access. This paper assumes familiarity with the basics of quantum computing. For convenience, we summarise several relevant definitions in the Appendix.

## 2 RELATED WORK

The two main paradigms for quantum computing are *gate-based* QC and adiabatic quantum computing (AQC). Our method uses quantum annealing, which is derived from AQC, and is *not* gate-based. The predominantly theoretical field of quantum machine learning (QML) investigates how quantum computations can be integrated into machine learning (Biamonte et al., 2016; Dunjko & Briegel, 2018; Sim et al., 2019; Havlíček et al., 2019; Du et al., 2020; Mariella & Simonetto, 2021; Kübler et al., 2021). Many QML methods assume gate-based quantum computers and define a quantum

variational layer, *i.e.,* a sequence of parametrised unitary transformations meant for execution on quantum hardware (Mitarai et al., 2018). QML methods are often optimised using variants of the backpropagation algorithm (McClean et al., 2018; Verdon et al., 2019; Beer et al., 2020). Quantum variational models were recently applied to combinatorial optimisation (Khairy et al., 2020) and reinforcement learning (Dunjko et al., 2016; Lockwood & Si, 2020). Instead of learning to regress unitary transformation parameters for gate-based QC, we learn to regress QUBO forms for QA.

In contrast to gate-based machines, QAs can already solve real problems formulated as QUBOs (Neukart et al., 2017; Teplukhin et al., 2019; Stollenwerk et al., 2019; Orus et al., 2019; Mato et al., 2021; Speziali et al., 2021). Recently, QCV has rapidly transitioned from theoretical considerations (Venegas-Andraca & Bose, 2003; Chin et al., 2020; Neven et al., 2008b;a) to practical algorithms leveraging quantum-mechanical effects of quantum computers, ranging from image retrieval and processing (Venegas-Andraca & Bose, 2003; Yan et al., 2016), classification (Boyda et al., 2017; Nguyen & Kenyon, 2019; Cavallaro et al., 2020; Willsch et al., 2020; Dema et al., 2020) and tracking (Li & Ghosh, 2020; Zaech et al., 2022), to problems on graphs (Zick et al., 2015; Seelbach Benkner et al., 2020; Mariella & Simonetto, 2021), consensus maximisation (Doan et al., 2022), shape alignment Noormandipour & Wang (2021); Seelbach Benkner et al. (2021), segmentation (Arrigoni et al., 2022) and ensuring cycle-consistency (Birdal et al., 2021; Yurtsever et al., 2022).

Many of these methods are evaluated on real quantum hardware, as both gate-based and QA machines can be accessed remotely (D-Wave Systems, 2022; Rigetti Computing, 2022).

We demonstrate the efficacy of our QuAnt approach on the applications of graph matching and point set alignment where we compare against recent quantum state-of-the-art methods Seelbach Benkner et al. (2020); Golyanik & Theobalt (2020), respectively.

Another line of work in different domains concerns learning the best adiabatic quantum algorithm. While some works (Pastorello et al., 2021; Pastorello & Blanzieri, 2019) develop an algorithm inspired by tabu search, our method uses a neural network to output a coupling matrix. Orthogonal to our work, others train neural networks to solve problem-specific QUBOs (Gabor et al., 2020). Nüßlein et al. (2022) optimize a blackbox function by finding a QUBO as surrogate model. QML on gate-based QC has been studied at length, but machine learning with QA remains largely underexplored, with only a few exceptions (*e.g.,* linear regression (Date & Potok, 2021) and binary neural networks (Sasdelli & Chin, 2021)). In stark contrast to existing QCV methods with analytically derived QUBOs (Li & Ghosh, 2020; Seelbach Benkner et al., 2020; 2021; Birdal et al., 2021), our approach enables more flexible and compact solution encodings.

QuAnt is also related to recent non-quantum approaches that aim to improve combinatorial optimisation by seamlessly integrating deep learning and combinatorial building blocks as custom layers and backpropagating through them (Ferber et al., 2020; Rolínek et al., 2020; Vlastelica et al., 2020). In this respect, ours is the first work that uses a quantum QUBO solver in neural architectures.

## 3 QUBO LEARNING APPROACH

We present a new meta-learning approach for regressing quadratic unconstrained binary optimisation problems (QUBOs) suitable for modern quantum annealers (QA); see Fig. 1. While existing works analytically derive QUBOs for different problems (Birdal et al., 2021; Seelbach Benkner et al., 2020; 2021; Zaech et al., 2022), we propose to instead *learn* a function that turns a problem instance into a QUBO to be solved by a QA. Specifically, we train a multi-layer perceptron (MLP) that takes a vectorised problem instance and regresses the QUBO weights such that the QUBO minimiser is the solution to the problem. Note that we only specify the bit encoding of the solution but let the network learn to derive QUBOs. Crucially, we show how training the MLP is possible despite quantum annealing (like any QUBO solver) being discontinuous and non-differentiable.

### 3.1 QUBOS AND QUANTUM ANNEALING

Quantum annealing is a metaheuristic to solve $\mathcal{NP}$-hard problems of the form $\arg\min_{\mathbf{s}\in\{-1,1\}^n} \mathbf{s}^\top\mathbf{J}\mathbf{s} + \mathbf{b}^\top\mathbf{s}$, where $\mathbf{s}$ is a binary vector, $\mathbf{J} \in \mathbb{R}^{n\times n}$ is a matrix of *couplings*, and $\mathbf{b} \in \mathbb{R}^n$ contains *biases* (McGeoch, 2014). We can rewrite this as a QUBO: $\arg\min_{\mathbf{x}\in\{0,1\}^n} \mathbf{x}^\top\mathbf{A}\mathbf{x}$, by substituting $\mathbf{x} = \frac{1}{2}(\mathbf{s}+\mathbb{1}_n)$ and $\mathbf{A} = \frac{1}{4}\mathbf{J}+\frac{1}{2}(\mathbf{J}\mathbb{1}_n+\mathbb{1}_n^\top\mathbf{J})+\frac{1}{2}\text{diag}(\mathbf{b})$.

In quantum annealing, the binary $n$-dimensional vector $\mathbf{x}$ describes the measurement outcomes of $n$ qubits. Annealing starts out with an equal superposition quantum state of the qubits that assigns an equal probability to all possible binary states $\{0, 1\}^n$. During the anneal, the couplings and biases of $\mathbf{A}$ are gradually imposed on the qubits. The adiabatic theorem (Born & Fock, 1928) implies that doing so sufficiently slow forces the qubits into a quantum state that assigns nonvanishing probability only to binary states that minimise the QUBO (Farhi et al., 2001). We then only need to measure the qubits to determine their binary state, which is our solution $\mathbf{x}$. For a more detailed description of quantum annealing, we refer to prior computer-vision works (Seelbach Benkner et al., 2020; 2021; Golyanik & Theobalt, 2020; Li & Ghosh, 2020) and Appendix A.

## 3.2 Network Architecture and Losses

In this section, we describe the network architecture that takes as input a problem instance and regresses a QUBO whose solution (*e.g.,* obtained via quantum annealing) solves the problem instance. For a given problem type, we require a problem description that is amenable to QUBO learning: A parametrisation of problem instances as real-valued vectors $\mathbf{p} \in \mathbb{R}^m$, and a parametrisation of solutions as binary vectors $\mathbf{x} \in \{0, 1\}^n$. Since we use supervised training, we additionally need a training set $\mathcal{D} = \{(\mathbf{p}_d, \hat{\mathbf{x}}_d)\}_{d=1}^D$ containing $D$ problem instances $\mathbf{p}_d$ with ground-truth solutions $\hat{\mathbf{x}}_d$.

We use a multilayer perceptron (MLP) with $L$ layers and $H$ hidden dimensions, ReLU activations (except for the last layer, which uses $\sin$ (Sitzmann et al., 2020)), and concatenating skip connections from the input into odd-numbered layers (except for the first and last layers). The input to the network is a problem instance $\mathbf{p}$, and the output is a QUBO matrix $\mathbf{A}$: $\mathbf{A} = \text{MLP}(\mathbf{p})$.

We could now use a dataset of problem instances and corresponding $\mathbf{A}$ to supervise the MLP directly. However, this requires specifying how $\mathbf{A}$ is to be derived for a certain instance, which comes with two downsides: (1) A problem-type-specific algorithm for analytically deriving instance-specific $\mathbf{A}$ needs to be designed to generate enough training data $\{(\mathbf{p}_d, \mathbf{A}_d)\}_{d=1}^D$, which is non-trivial, and (2) The binary parametrisation ($\mathbf{x}$) of the solution space depends on the algorithm, which can lead to more variables than intrinsically needed by the problem (e.g., if $\mathbf{x}$ needs to represent one of $k$ numbers, a one-hot parametrisation would have length $n = k$, while a binary-encoding parametrisation would have length $n = \log k$). This is particularly problematic as contemporary quantum hardware only provides a limited number of qubits. We thus choose to supervise $\mathbf{A}$ not directly and, instead, supervise the solutions of the QUBO. This strategy tackles both issues as it lets the network *learn* an algorithm *compatible* with the (potentially shorter) solution parametrisation. Therefore, our method is easily applicable to new problem types, as we show in Secs. 3.4 and 3.5.

The regressed $\mathbf{A}$ defines a QUBO, which can be solved by any QUBO solver. But how can we, during training, backpropagate gradients from the solution binary vector $\mathbf{x}$ through the QUBO solver despite these solvers having zero gradients almost everywhere (Vlastelica et al., 2020)? We circumvent this issue by exploiting a contrastive loss (*cf.* LeCun et al. (2006, Eq. (10))) as follows: We know the energy of the ground-truth solution $\hat{\mathbf{x}}$ of the problem instance, namely $\hat{\mathbf{x}}^\top \mathbf{A} \hat{\mathbf{x}}$. If the energy of the minimiser $\mathbf{x}^* = \mathbf{x}^*(\mathbf{A})$ of the current QUBO is lower, then $\mathbf{A}$ does not yet describe a QUBO that outputs the right solution. We, therefore, seek to push the energy of $\hat{\mathbf{x}}$ lower while pulling the energy of the minimiser $\mathbf{x}^*$ up:

$$L_{\text{gap}} = \hat{\mathbf{x}}^\top \mathbf{A} \hat{\mathbf{x}} - \mathbf{x}^{*\top} \mathbf{A} \mathbf{x}^*, \tag{1}$$

which has a zero gradient if $\mathbf{x}^* = \hat{\mathbf{x}}$, as desired. $L_{\text{gap}}$ avoids backpropagation through $\mathbf{x}^*$ and is even compatible with automatic differentiation. It yields the following update for an entry of $\mathbf{A}$, which can then be further backpropagated into the MLP via the chain rule:

$$\frac{\partial L_{\text{gap}}(\mathbf{A})}{\partial \mathbf{A}_{i,j}} = 2\hat{\mathbf{x}}_i \hat{\mathbf{x}}_j - 2\mathbf{x}_i^* \mathbf{x}_j^* - 2 \frac{\partial \mathbf{x}^*(\mathbf{A})}{\partial \mathbf{A}_{i,j}} \mathbf{A} \mathbf{x}^*, \tag{2}$$

where the last term comes from the dependency of $\mathbf{x}^*$ on $\mathbf{A}$ via the QUBO solver. While intuitively useful, we note that this term is zero "almost everywhere" (in the mathematical sense), and we hence ignore it as it provides almost no information. Note that this is a common approximation (*e.g.,* auto-differentiation frameworks backpropagate through $\max(\cdot)$ pooling in the same manner).

Unfortunately, $L_{\text{gap}}$ alone would not prevent degenerate $\mathbf{A}$, which have multiple solutions, including undesirable ones. We, therefore, discourage such $\mathbf{A}$ that have more than one solution $\mathbf{x}^*$:

$$L_{\text{unique}} = -|\hat{\mathbf{x}}^\top \mathbf{A} \hat{\mathbf{x}} - \mathbf{x}^{+\top} \mathbf{A} \mathbf{x}^+|, \tag{3}$$

where $\mathbf{x}^+$ is the $\mathbf{x}$ that minimises $\mathbf{x}^\top \mathbf{A}\mathbf{x}$ over $\{0,1\}^n \setminus \{\mathbf{x}^*\}$, and $|\cdot|$ is the absolute value operator.

In addition to these data terms, we found it helpful to regularise the network by encouraging sparsity on all intermediate features of the MLP (Liu et al., 2015):

$$L_{\mathrm{mlp}} = \sum_{\mathbf{f} \in \mathcal{F}} \frac{1}{|\mathbf{f}|} \|\mathbf{f}\|_1, \tag{4}$$

where $\mathcal{F}$ is the set of all network layer outputs (except for the last layer). The total loss then reads:

$$L = L_{\mathrm{gap}} + \lambda_{\mathrm{unique}} L_{\mathrm{unique}} + \lambda_{\mathrm{mlp}} L_{\mathrm{mlp}}, \tag{5}$$

where we set $\lambda_{\mathrm{mlp}} = 10^{-4}$ and $\lambda_{\mathrm{unique}} = 10^{-3}$ regardless of problem type.

## 3.3 QUBOs on D-Wave Quantum Annealers

We can use D-Wave to solve any generated QUBO $\mathbf{A}$, where $\mathbf{A}_{i,j} \in \mathbb{R}$ describes the direction and coupling strength between logical qubits $i$ and $j$. However, each physical qubit in the annealer is only connected to a small subset of other physical qubits, which makes the regressed $\mathbf{A}$ not directly compatible with the annealer. We tackle this issue by manually pre-determining a sparse connectivity pattern of the physical qubits and then masking out the other entries of $\mathbf{A}$ before solving, such that the MLP focuses on only these sparse entries. For example, when $\mathbf{x} \in \mathbb{R}^n$ with $n{=}8$, we can use D-Wave's Chimera architecture, which is made up of interconnected $\mathcal{K}_{4,4}$ unit cells (Dattani et al., 2019). ($\mathcal{K}_{4,4}$ is a complete bipartite graph with two sets of four qubits each.) Since $n{=}8$, we can fit one problem instance into one unit cell, which allows us to anneal many problem instances in parallel by putting them in different unit cells. This speeds up training and saves expensive time on the quantum annealer. For larger problems with $n{=}32$, we can use D-Wave's Pegasus architecture (Boothby et al., 2020), which has more interconnections (qubit couplings) between its $\mathcal{K}_{4,4}$ unit cells than Chimera. We use four such unit cells per problem instance, following D-Wave's pattern (Til). See Fig. 4b for an exemplary colour-coded qubit connectivity pattern of $\mathbf{A}$.

Given our full method, we next show how it can be applied to three problem types, *i.e.,* graph matching, point set registration, and rotation estimation; see Appendix for further details.

## 3.4 Graph Matching

The goal of graph matching is to determine correspondences from $k$ key points in two images or graphs; see Fig. 2 for an example. This can be formalised as a quadratic assignment problem with a permutation matrix representing the matching (Seelbach Benkner et al., 2020): $\arg\min_{\mathbf{X} \in \mathbb{P}_k} \mathbf{x}^\top \mathbf{W} \mathbf{x}$, where $\mathbb{P}_k$ is the set of permutation matrices, $\mathbf{x} = \mathrm{vec}(\mathbf{X}) \in \{0,1\}^{k^2}$ is the vectorised permutation matrix, and $\mathbf{W} \in \mathbb{R}^{k^2 \times k^2}$ contains pairwise weights. Unfortunately, the permutation constraint cannot be directly realised on the quantum annealer.

Instead, note that a permutation $P : [k] \to [k]$ is fully defined by the sequence $P(1), P(2), \ldots, P(k)$. Our method can use an efficient binary encoding for each entry of this sequence, using only $k \log k$ binary variables in total. Note that not all vectors $\mathbf{x} \in \{0,1\}^{k \log k}$ are valid permutations. As an optional post-processing step, we can perform a *projection* to the nearest permutation with respect to the Hamming distance in our binary encoding. Unless stated otherwise, we do not apply this post-processing.

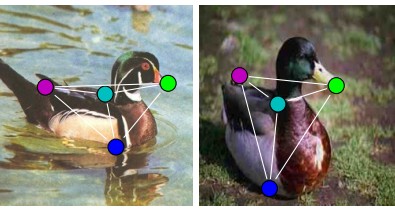

(a) Source      (b) Target

Figure 2: Example matching on four key points from the Willow dataset (Cho et al., 2013). Corresponding points (same colours) are found based on feature similarity.

In addition to the solution parametrisation, we also need to design the problem description $\mathbf{p}$. For real data, we use $\mathbf{p} = \mathrm{vec}(\mathbf{W})$, where the diagonal contains cosine similarities between the feature vectors extracted with AlexNet (Krizhevsky et al., 2012) pre-trained on ImageNet (Deng et al., 2009) of all key point pairs, and the off-diagonal follows the geometric term from (Torresani et al., 2008, Eq. (6)). For evaluation, we also introduce the synthetic *RandGraph* dataset; it uses matrices of random distances $\mathbf{D} \in \mathbb{R}^{k \times k}$ with entries $\mathbf{D}_{i,j} \in \mathcal{U}(0,1)$ to define $\mathbf{W}_{k \cdot i + P(i), k \cdot j + P(j)} = |\mathbf{D}_{i,j} - \mathbf{D}_{P(i),P(j)}|$. The MLP thus learns to compress the input matrix into a much smaller QUBO.

### 3.5  Point Set Registration and Rotation Estimation

In 2D point set registration, we are given two point sets with potentially different numbers of points and no correspondences, and we seek to find a rotation angle that best aligns them. We follow Golyanik *et al.* (Golyanik & Theobalt, 2020) and use the vectorised form of their input matrix to represent a problem instance. We parametrise the solution space of $\mathbf{x} \in \{0,1\}^9$ by splitting the output space $[0, \frac{1}{3}\pi]$ into $2^9$ equally sized bins and consecutively indexing them with a 9-bit integer.

In 3D rotation estimation, we are given two 3D point clouds with known matches and seek to estimate the 3D rotation aligning them. We represent a problem instance by the vectorised covariance matrix of the two point clouds; 3D rotation is parametrised by Euler angles $\alpha, \beta, \gamma$. We discretise each angle into $2^5$ bins, such that $\mathbf{x} \in \{0,1\}^{15}$.

## 4  Experimental Evaluation

We next experimentally evaluate QuAnt. Our goal is to show that it outperforms the previous quantum state of the art. For reference, we also report comparisons against specialised classical methods.

**Data.** We evaluate graph matching on the Willow object dataset (Cho et al., 2013), which contains labelled key points. We use $k{=}4$ randomly chosen key point pairs per image. We use 5640 images for training, and test on 846 images. Both of the sets are obtained via `pygmtools` (ThinkLab, 2021). We also evaluate on our synthetic dataset *RandGraph* (see Sec. 3.4), with both $k{=}4$ and $k{=}5$. We evaluate 2D point set registration on the 2D Shape Structure dataset (Carlier et al., 2016) providing 2D silhouette images of real-world objects. We treat the silhouette outlines as 2D points. We use 500 shapes from various classes for training, and test on 50 shapes. For each shape, we apply 1000 (for train) or 100 (for test) different rotations of up to $60°$ and pick random pairs to generate problem instances. For 3D rotation estimation, we evaluate on ModelNet10 (Wu et al., 2015), which contains CAD models of ten object categories. We proceed with point cloud representations of each shape. We use 300 shapes from various classes for training, and test on 30 shapes from various classes. For each shape, we apply 1000 different 3D rotations with angle ranges $\alpha, \gamma \in [-\frac{1}{9}\pi, \frac{1}{9}\pi]$ and $\beta \in [-\frac{1}{18}\pi, \frac{1}{18}\pi]$ and pick random pairs to generate problem instances.

**Comparisons.** We compare QuAnt to two baselines and specialised methods, depending on the problem type. For all problem types, we demonstrate the power of using QUBOs compared to the *Diag* baseline that regresses a diagonal QUBO matrix $\mathbf{A}$ (which is trivially solvable). While this baseline ablates the QUBO itself, we also consider a more natural neural network baseline, *i.e.,* *Pure*, that regresses the binary solution directly (there is no activation after the last layer) and uses an $\ell_1$-loss between the output and $\hat{\mathbf{x}}$ instead of $L_{\text{gap}}$ and $L_{\text{unique}}$. At test time, we threshold the network output of Pure at 0 to obtain binary vectors. For QuAnt, Diag and Pure variants, we experiment with all combinations of the numbers of layers $L \in \{3, 5\}$ and hidden dimensions $H \in \{32, 78\}$.

We compare our graph matching results with the *Direct* baseline on Willow (Cho et al., 2013); we directly solve the quadratic assignment problem given by $\mathbf{W}$ with exhaustive search, which provides an upper bound for our method. We also compare against Quantum Graph Matching (QGM) (Seelbach Benkner et al., 2020), to which we pass our input matrices $\mathbf{W}$. For 2D point set registration, we compare against the analytic quantum method (AQM) (Golyanik & Theobalt, 2020), which is an upper bound for our technique since we take its vectorised QUBO as input, and against the classical, specialised ICP algorithm (Lu & Milios, 1997). For 3D rotation estimation, we use Procrustes as a classical specialised method, which is thus an upper bound for our (general) method.

**Metrics.** We measure accuracy of the graph matching solutions as the percentage of correctly recovered permutation matrices. For 2D point set registration and 3D rotation estimation, we quantify the difference between the known ground-truth rotations and the estimated rotations by their geodesic distances (angles) in the rotation groups $SO(2)$ and $SO(3)$, respectively.

**QUBO Solvers.** For graph matching, we follow Sec. 3.3 to make our regressed QUBOs compatible with the QA. Due to a restricted QA compute budget, we train and test with simulated annealing unless stated otherwise. For the point cloud experiments, we regress dense $\mathbf{A}$ and use our exhaustive search implementation at train and test time unless stated otherwise. When evaluating on the QA, we rely on minor embeddings to make the regressed $\mathbf{A}$ compatible with the QA. Please refer to the Appendix for the details.

## 4.1 RESULTS

**General Baselines.** The quantitative results for graph matching, point set registration, and rotation estimation are reported in Tables 1, 2, and 3, respectively. Across network sizes and all three problem types, the results show that having a full, $\mathcal{NP}$-hard QUBO (ours) instead of only a diagonal QUBO (Diag) is advantageous. We also find that the proposed method yields better results than Pure on both point set registration and rotation estimation, although Pure yields better results for graph matching.

Table 1: Comparison to general baselines on graph matching. We report the accuracy (in %).

(a) RandGraph for $k=4$.

|  | **Ours** | Diag | Pure |
|---|---|---|---|
| $L=3, H=32$ | 9 | 8 | **91** |
| $L=3, H=78$ | 30 | 18 | **96** |
| $L=5, H=32$ | 11 | 11 | **89** |
| $L=5, H=78$ | 49 | 43 | **96** |

(b) Willow object dataset (Cho et al., 2013) for $k=4$. Trained for 300 epochs with $L=5, H=78$.

| **Ours** | Diag | Pure | Direct |
|---|---|---|---|
| 69 | 53 | 90 | **97** |

Table 2: Comparison to general baselines on point set registration. We report averages of the mean errors and their standard deviations over three runs.

|  | **Ours** | Diag | Pure |
|---|---|---|---|
| $L=3, H=32$ | $8.4 \pm 0.8$ | $11.1 \pm 1.3$ | $\mathbf{8.2 \pm 1.2}$ |
| $L=3, H=78$ | $\mathbf{7.2 \pm 1.1}$ | $8.3 \pm 0.7$ | $9.3 \pm 1.9$ |
| $L=5, H=32$ | $\mathbf{8.6 \pm 0.5}$ | $10.9 \pm 1.2$ | $9.3 \pm 1.9$ |
| $L=5, H=78$ | $\mathbf{6.8 \pm 0.3}$ | $7.7 \pm 0.5$ | $11.3 \pm 4.5$ |

Table 3: Comparison to general baselines on rotation estimation. We report averages of the mean errors and their standard deviations over three runs.

|  | **Ours** | Diag | Pure |
|---|---|---|---|
| $L=3, H=32$ | $5.9 \pm 3.0$ | $\mathbf{5.4 \pm 1.0}$ | $7.9 \pm 0.5$ |
| $L=3, H=78$ | $\mathbf{4.1 \pm 0.5}$ | $5.0 \pm 0.3$ | $7.1 \pm 0.1$ |
| $L=5, H=32$ | $\mathbf{3.7 \pm 0.8}$ | $5.0 \pm 0.4$ | $16.2 \pm 7.1$ |
| $L=5, H=78$ | $\mathbf{3.4 \pm 0.4}$ | $4.7 \pm 0.2$ | $10.1 \pm 1.8$ |

**Specialised Methods.** For reference, we compare QuAnt to methods specialised to a certain problem type. Since our approach is general, they mostly provide an upper bound for our performance.

We evaluate QGM (Seelbach Benkner et al., 2020) on several RandGraph instances. We confirm their finding that strongly enforcing permutation constraints eventually retrieves the right permutation as the sample with the lowest energy. However, using the analytical bound for the penalty term leads to a success probability (*i.e.,* the probability of getting the best solution across anneals) smaller than random guessing due to experimental errors in the couplings. Next, we find that the QUBOs of QuAnt are much smaller and better suited to be solved with a quantum annealer than QGM's. For RandGraph with $k=5$, our method needs 15 physical qubits while their baseline and row-wise methods need 89 qubits on average and a chain length of four, and their *Inserted* method needs, on average, 39 qubits and a chain length of three on D-Wave Advantage. Thus, our success probability of $26\%$ when evaluating on test data is orders of magnitude higher than *Inserted*'s $0.22\%$ (best in QGM). This shows how QuAnt improves over the quantum state of the art even though we merely focus on the solution with the lowest energy across anneals, while they focus on the success probabilities. We refer to Appendix D for a detailed evaluation. Table 1b confirms that *Direct* is an upper bound to our approach.

Table 8 shows quantitative results for 2D point set matching without noise. AQM slightly outperforms QuAnt, which is expected as we take AQM's QUBO as input; hence, its performance is an upper bound for our method. Fig. 3 shows a qualitative example.

As expected, quantitative results in Table 7 (with no incorrect correspondences) show that classical, specialised Procrustes performs better than our general method on 3D rotation estimation. Note that our technique yields better results than Procrustes under test-time noise, as we discuss later in detail.

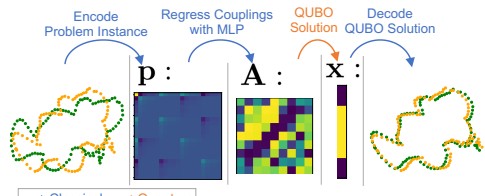

Figure 3: Test-time example inputs and outputs of QuAnt trained for 2D point set registration.

## 4.2 ABLATION

We ablate $L_{\text{unique}}$ and $L_{\text{mlp}}$ in Table 4. For graph matching, we use RandGraph with $k=4$. We find that removing either $L_{\text{unique}}$ or $L_{\text{mlp}}$ leads to mixed results on graph matching and worse results in almost all cases for points set registration and rotation estimation.

Table 4: Loss ablations. We report accuracy for graph matching (in %) and mean/median error otherwise.

|  | Graph Matching | | | Point Set Registration | | | Rotation Estimation | | |
|---|---|---|---|---|---|---|---|---|---|
|  | w/o $L_{\text{unique}}$ | w/o $L_{\text{MLP}}$ | Ours | w/o $L_{\text{unique}}$ | w/o $L_{\text{MLP}}$ | Ours | w/o $L_{\text{unique}}$ | w/o $L_{\text{MLP}}$ | Ours |
| $L=3, H=32$ | 7 | **9** | **9** | 17.8 / 12.0 | 18.6 / 13.4 | **15.0 / 8.7** | 5.1 / 5.0 | 3.4 / 3.0 | **3.4 / 3.0** |
| $L=3, H=78$ | **30** | 29 | **30** | 15.5 / 8.0 | 21.8 / 17.0 | **14.5 / 7.7** | **2.9 / 3.0** | 4.6 / 5.0 | 4.2 / 4.0 |
| $L=5, H=32$ | **14** | 6 | 11 | 18.1 / 11.7 | 19.0 / 7.7 | **9.0 / 4.6** | 3.4 / 3.0 | 2.5 / 2.0 | **2.3 / 2.0** |
| $L=5, H=78$ | 46 | **54** | 49 | 18.3 / 11.7 | **17.8** / 11.7 | 18.5 / **7.7** | 3.7 / 4.0 | 4.1 / 4.0 | **3.3 / 3.0** |

## 4.3 EVALUATION ON D-WAVE

By design, our QuAnt is agnostic to the type of QUBO solver used. After training with exhaustive search, we compare how the performance on the test set differs under exhaustive search, SA, or QA. The results in Fig. 4a show that the exact solutions of exhaustive search only slightly outperform the less computationally expensive QA and SA. Moreover SA yields results very similar to QA.

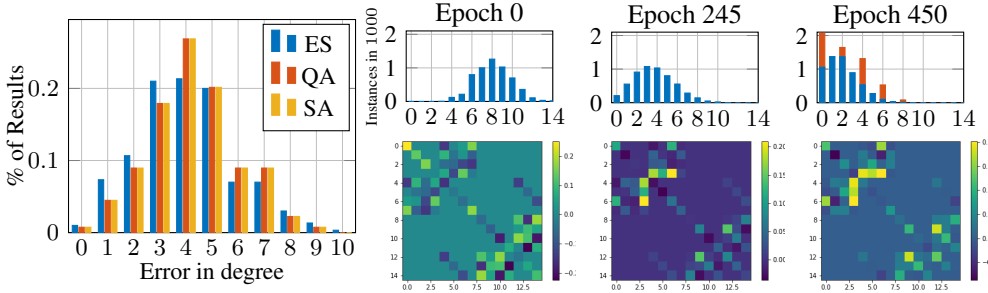

(a) Errors on rotation estimation.          (b) Evolution of result (top) and coupling matrix (bottom).

Figure 4: (a) Histogram of errors for rotation estimation using exhaustive search (ES), quantum annealing (QA), and simulated annealing (SA) at test time. The maximum error on the x-axis amounts to 59 and no methods have higher errors than shown. (b) Evolution over different epochs of the Hamming distance between predicted solutions and the ground truth (top), and coupling matrix when training our approach for graph matching (bottom). (Top): The x-axis shows the Hamming distance. Blue indicates unprojected results, and red means after projection to a permutation. We only project after training.

Next, we compare the test-time results of QA and SA (after training the method with the same technique, QA and SA, respectively). See Table 5 for the results, both with and without projecting the final binary solution to a valid permutation encoding during post-processing. Training with QA delivers better results than training with SA. We attribute this to a better second-best solution $\mathbf{x}^+$ used by $L_{\text{unique}}$. While SA yields solutions $\mathbf{x}^*$ that are comparable to QA,

Table 5: QuAnt ($L=5, H=78$) on RandGraph ($k=5$) trained for 450 epochs (QA or SA).

|  | SA | QA |
|---|---|---|
| Before projection | 10 | **18** |
| After projection | 24 | **36** |

its second-best solutions are worse than QA's. We refer to the appendix for details. Unfortunately, real-world compute resources for training with QA remain limited, as of this writing. We, therefore, fall back on SA for larger-scale experiments in this work. However, Table 5 suggests that our results could improve noticeably on QA.

## 4.4 FURTHER ANALYSIS

**Training.** Fig. 4b visualises how the instantaneous solution and $\mathbf{A}$ matrix evolve for graph matching.

**Varying Problem Difficulty.** We provide a more detailed analysis of the performance of our method on point set registration for varying difficulty levels. Table 6 shows that a larger input misalignment between the two point clouds worsens the results, as expected.

Table 6: Interval analysis for point set registration with $L=5, H=78$. We evaluate on point cloud pairs with ground-truth angles uniformly sampled within the given intervals. We report the mean/median error in degrees.

| angle interval | $0 - 1\frac{\pi}{18}$ | $1\frac{\pi}{18} - 2\frac{\pi}{18}$ | $2\frac{\pi}{18} - 3\frac{\pi}{18}$ | $3\frac{\pi}{18} - 4\frac{\pi}{18}$ | $4\frac{\pi}{18} - 5\frac{\pi}{18}$ | $5\frac{\pi}{18} - 6\frac{\pi}{18}$ |
|---|---|---|---|---|---|---|
| mean / median | 4.0 / 1.9 | 5.1 / 2.6 | 6.4 / 3.0 | 7.9 / 3.8 | 7.9 / 3.8 | 8.4 / 4.0 |

**Robustness to Noise.** We investigate the robustness of our method and other approaches against input noise at test time after training without noisy data. We look at rotation estimation, where we randomly pick a fixed percentage of points and randomly permute their correspondences (among themselves). Table 7 contains results. The quality of QuAnt's results barely degrades with increasing noise levels, even for $20\%$ of incorrect correspondences. QuAnt already outperforms the classical Procrustes for even $1\%$ of incorrect correspondences, even though Procrustes also starts from the same covariance matrix. We observe that the advantage of our method grows with larger noise levels. QuAnt also consistently performs better than the general baselines, which are similarly robust to increasing noise levels.

Table 7: Robustness to varying amounts of incorrect test-time correspondences in rotation estimation. We report mean/median error for $L=3$, $H=32$. The first column specifies the percentage of incorrect correspondences at test time.

| % | **Ours** | Procrustes | Diag | Pure |
|---|---|---|---|---|
| 0 | 3.9 / 4.0 | **0.0 / 0.0** | 5.6 / 6.0 | 8.1 / 8.0 |
| 1 | **3.4 / 3.0** | 5.8 / 3.0 | 5.7 / 6.0 | 8.2 / 8.0 |
| 5 | **3.4 / 3.0** | 25.7 / 13.0 | 6.0 / 6.0 | 8.2 / 8.0 |
| 10 | **3.2 / 3.0** | 43.8 / 21.0 | 6.2 / 6.0 | 8.2 / 8.0 |
| 15 | **3.5 / 3.0** | 64.7 / 58.0 | 6.2 / 6.0 | 8.2 / 8.0 |
| 20 | **3.7 / 3.0** | 75.3 / 79.0 | 5.8 / 6.0 | 8.2 / 8.0 |

We next look at point set registration under input noise at test time and after training without noise. Here, we add uniform noise to one point cloud, where the range of the noise is a percentage of the maximum extent of the point cloud. Table 8 contains the results. ICP, an iterative approach, is robust to the noise, gives highly accurate results and, thus, outperforms the competing non-iterative approaches. Since QuAnt takes as input the vectorised QUBO that AQM solves, AQM constitutes an upper bound for the performance of our approach. However, QuAnt could, in principle, scale to 3D point set matching while AQM's solution parametrisation severely inhibits scaling to larger problems. Finally, QuAnt performs better than the general baselines.

Table 8: Robustness to varying amounts of uniform noise in point set registration. We report mean/median error for $L=5, H=78$ and the number of logical/physical qubits. The first column states the range of the noise in % of the maximum extent of the point cloud. "†": uncoupled qubits.

| % | **Ours** | Diag | Pure | AQM |
|---|---|---|---|---|
| 0 | 5.8 / 3.5 | 7.3 / 4.7 | 6.8 / 5.9 | **4.3 / 2.6** |
| 5 | 6.4 / 3.3 | 7.0 / 5.2 | 7.0 / 6.1 | **4.5 / 2.9** |
| 10 | 6.5 / **3.3** | 8.4 / 5.2 | 7.1 / 6.5 | **5.6** / 3.8 |
| 15 | 7.2 / **3.5** | 9.5 / 5.9 | 7.9 / 6.7 | **5.6** / 3.8 |
| 20 | 10.3 / 5.4 | 11.6 / 6.6 | 8.2 / 6.8 | **5.9 / 3.3** |
| Qubits | **9/14** | $(9/9)^\dagger$ | n/a | $21/\sim55$ |

## 5 DISCUSSION

**Limitations and Future Work.** As all learning-based approaches, QuAnt can perform worse on problem instances that fall significantly outside the training distribution. While our general method does not outperform classical methods specialised on certain problem types, we achieve performance on par with hand-crafted QUBO designs used in state-of-the-art QCV methods. We achieve this while greatly reducing the effort required for new problem types. For our point cloud experiments, we rely on minor embeddings to transfer the regressed dense QUBOs to the QA. On existing hardware, large minor embeddings can worsen the resulting quality noticeably. However, we only need to embed a QUBO with nine logical qubits into 14 physical qubits. Although our focus is on a general design, our core idea of learning QUBOs can be specialised to any given problem type by designing a more specific network architecture and losses that capture priors for the problem type.

**Conclusion.** We showed that learning to regress QUBO forms for different problems instead of deriving them analytically can be a reasonable alternative to existing methods. We showed the generality of QuAnt on diverse problem types. Our experiments demonstrated that learning QUBO forms and solving them either on a quantum annealer or with simulated annealing, in most cases, leads to better results than directly regressing solutions. Moreover, QuAnt significantly outperformed the previous quantum state of the art in graph matching and rotation estimation in the setting with noise. We believe our work considerably broadens the available toolbox for development and analysis of quantum computer vision methods and opens up numerous avenues for future research.

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

## APPENDIX

This appendix provides more details on adiabatic quantum computing in Sec. A. We provide training and implementation settings in Sec. B. Further details on the problem description for graph matching and a failure case are in Sec. C. Sec. D contains a deeper comparison with QGM (Seelbach Benkner et al., 2020) and Sec. E compares SA and QA. In Sec. F, we provide further quantitative and qualitative results on rotation estimation. Finally, Sec. G contains more details and experiments on point set registration.

## A    QUANTUM COMPUTING BACKGROUND

### A.1    QUANTUM ANNEALING IN DETAIL

As we have seen, quantum annealing is a metaheuristic to solve the $\mathcal{NP}$-hard Ising problem:

$$\arg\min_{\mathbf{s}\in\{-1,1\}^n} \mathbf{s}^\top \mathbf{J}\mathbf{s} + \mathbf{b}^\top \mathbf{s}, \tag{6}$$

where $\mathbf{s}$ is a binary vector, $\mathbf{J} \in \mathbb{R}^{n\times n}$ is a matrix of *couplings*, and $\mathbf{b} \in \mathbb{R}^n$ contains *biases* (McGeoch, 2014). Here, we give a brief overview of how this fits in the framework of quantum mechanics. D-Wave quantum annealers rely on magnetic fluxes in superconducting Niobium loops (Orlando et al., 1999). The direction of the current flowing through them can be modelled as a qubit, *i.e.,* as a two-dimensional, complex, normalised vector $|\psi\rangle \in \mathbb{C}^2$ in the Dirac notation. In the so-called computational basis, the basis vectors correspond to the current flowing clockwise or anti-clockwise. After measuring the state, the system will collapse to either basis state. The absolute value of the complex-valued coefficients of the linear combination (probability amplitudes) is the probability of each outcome after measurement (*e.g.,* clockwise or anti-clockwise current). The state space of $n \in \mathbb{N}$ qubits can be expressed with the tensor product $\bigotimes_{i=1}^{n} \mathbb{C}^2$ and is thus a $2^n$-dimensional complex vector space. We need that many parameters because *entangled* states cannot be described separately. If the two states $|\psi\rangle, |\phi\rangle$ corresponding to different physical systems (*e.g.,* two niobium loops or two atoms) can be described independent from each other, the whole system is described by $|\psi\rangle \otimes |\phi\rangle$. Note that if every state could be decomposed this way, one would only need $2n$ parameters.

The evolution of a quantum state $|\psi\rangle$ over time can be described with the time-dependent Schrödinger equation:

$$H |\psi\rangle = i\hbar \frac{\partial}{\partial t} |\psi\rangle, \tag{7}$$

where the Hamilton operator $H$ is a Hermitian Matrix describing the possible energies of the system, $i$ is the imaginary unit, $\hbar$ is a constant, and $t$ denotes time. For adiabatic quantum computing, one needs a problem Hamiltonian $H_P$, where the eigenvector corresponding to the lowest eigenvalue is

a solution to the particular Ising problem, and an initial Hamiltonian $H_I$ with an easy-to-prepare ground state. The Adiabatic Theorem (Born & Fock, 1928) states that if we start with the ground state of $H_I$ and take a sufficiently long time $\tau$ to gradually change from $H_I$ to $H_P$, *e.g.,* with:

$$H(t) = (1 - \frac{t}{\tau})H_I + \frac{t}{\tau}H_P, \tag{8}$$

then we end up in the ground state of $H_P$. From the latter, we can deduce the solution of the particular Ising problem. Simulating this whole process classically can be difficult (or even intractable) because we are dealing with $2^n \times 2^n$ matrices $H_I$ and $H_P$, where $n$ is the number of qubits. How difficult the classical simulation is, depends on the exact form of the Hamiltonians. (Particularly promising for speed-ups are, *e.g.,* so-called non-stoquastic Hamiltonians (Albash & Lidar, 2018).)

## A.2 LOGICAL AND PHYSICAL QUBITS

The QUBO defines the couplings between two *logical* qubits $i$ and $j$. Such a QUBO can contain couplings between any two qubits. However, in contemporary hardware realisations, each *physical* qubit is only connected to a few others (see Fig. 5a and Fig. 5b). In the main paper, we show how the QUBO matrix **A** can account for this sparse connectivity pattern by setting entries between logical qubits $i$ and $j$ to 0 if the physical qubits $i$ and $j$ have no connection. Still, D-Wave supports denser connectivity patterns than what is implied by the hardware: Multiple physical qubits can be chained together to represent a single logical qubit of **x** that has many connections. The physical qubits in the chain will then have strong couplings along the chain to encourage them to all end up in the same final state (either all 0 or all 1), representing the final state of the corresponding logical qubit. This is formalised as a minor embedding (of the connectivity graph of the logical qubits) into the connectivity graph of the physical qubits. Using the heuristic method of Cai and colleagues (Cai et al., 2014) is popular to determine the minor embeddings in practice.

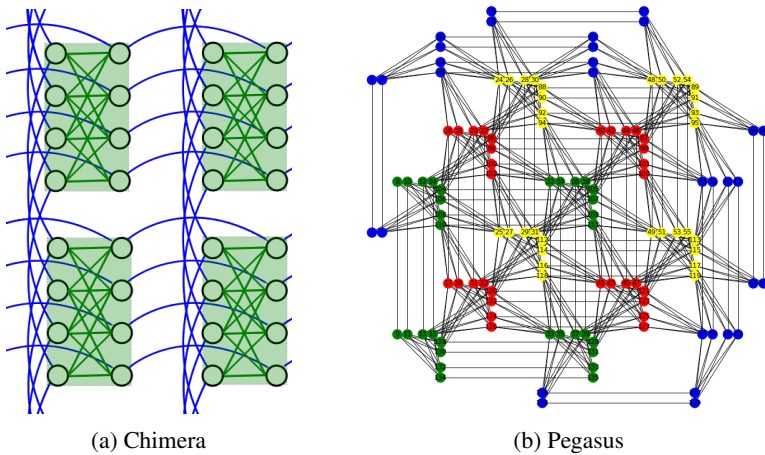

(a) Chimera                      (b) Pegasus

Figure 5: Visualisation of qubit connectivities. (a) The connectivity pattern of the physical qubits in the Chimera architecture. The unit cells (green boxes) have fewer interconnections than on Pegasus. (b) The connectivity pattern of the physical qubits in the Pegasus architecture. Green, red, and yellow correspond to one problem instance each. Images due to D-Wave (D-Wave Systems, Inc.).

## B IMPLEMENTATION DETAILS

Our code, which we will release, is implemented in Pytorch (Paszke et al., 2019). We use Adam (Kingma & Ba, 2014) with a learning rate of $10^{-3}$ for training. For graph matching on RandGraph with $k=4$, we use a batch size of 141 and train for 150 epochs, which takes about seven hours. For RandGraph $k=5$, we use 450 epochs, which takes about 23 hours. For Willow, we train for 300 epochs, which takes about 14 hours. The baselines are trained for the same number of epochs. While Diag takes a comparable amount of time, the Pure baseline takes about three minutes to train. For the experiments with $k=5$, we use $10^{-5}$ as the learning rate.

For the point cloud experiments, we use a batch size of 32 and train for 20 epochs, which takes about four hours. We train Pure and Diag with the same batch sizes and epochs. The training of the Diag baseline takes four hours as well, and Pure is trained within three hours. When solving a QUBO on a QA, we anneal 100 times and pick the lowest-energy solution. We access the QA via Leap 2 (D-Wave Systems, 2022) using the Ocean SDK (D-Wave Systems, Inc., 2022c). When solving with SA, we use 100 iterations from the default `neal` SA sampler.

## C  GRAPH MATCHING

### C.1  PROBLEM DESCRIPTION

Here, we describe the design of the problem description $\mathbf{p}$ for graph matching. We use $\mathbf{p} = \text{vec}(\mathbf{W})$, where the diagonal of $\mathbf{W}$ contains cosine similarities between the feature vectors extracted with AlexNet (Krizhevsky et al., 2012) pre-trained on ImageNet (Deng et al., 2009) of all pairs of key points. The off-diagonal follows the geometric term described in (Eq. (7)) from Torresani *et al.* (Torresani et al., 2008). In particular, we use the term $\mathbf{W}_{\text{geom}}$ from Eq. (7) from Torresani *et al.* (Torresani et al., 2008) with minus signs in the beginning and in the exponential, and set $\eta$=0.98. The convex combination with $\mathbf{W}_{\text{Alex}}$, where the cosine similarities of the feature vectors are on the diagonal, is then:

$$\mathbf{W} = \tau \mathbf{W}_{\text{Alex}} + (1 - \tau)\mathbf{W}_{\text{geom}}. \tag{9}$$

We choose $\tau$=0.81 and $\eta$=0.98 such that the QAP often coincides with the ground-truth correspondences (see Table 1b, "Direct" from the main paper).

### C.2  FAILURE CASE

We show a failure case of our method when applied to graph matching in Fig. 6. It occurs due to large differences in the observed appearance.

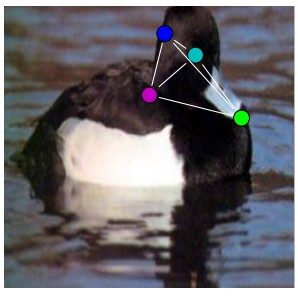 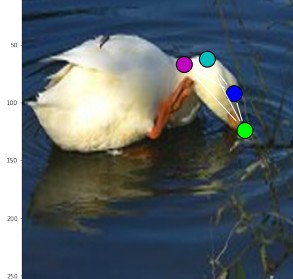

(a) Source                     (b) Matching

Figure 6: Failure case for graph matching. We visualise the ground truth of an image pair. Here, our method does not find the correct matching: Only the beak and neck are matched correctly, while the geometric information for the other key points differs too strongly.

## D  DETAILED COMPARISON WITH QGM (SEELBACH BENKNER ET AL., 2020)

Here, we compare our method with QGM (Seelbach Benkner et al., 2020) in detail. Note that the focus of both works differs. Their work focuses more on the probability distribution of the retrieved solutions. Our work is more concerned with incorporating the quantum annealer into the training pipeline. When training the neural network, $L_{\text{gap}}$ equation 1 uses the retrieved solution with the smallest energy across anneals, while they are also interested in the success probabilities, *i.e.,* the probability to get the best solution across anneals.

The *individual* QUBOs occurring in our QuAnt framework are much easier to solve by the QA than QUBOs that would arise in QGM (Seelbach Benkner et al., 2020). To show this, we compute the average success probabilities of the various methods from QGM (Seelbach Benkner et al., 2020)

Table 9: Average success probabilities of different QGM variants (Seelbach Benkner et al., 2020) over 141 problem instances on RandGraph compared to QuAnt with $k = 5$, in %.

| Inserted | Baseline | Row-wise | Ours |
|----------|----------|----------|------|
| 0.22 | 0.07 | 0.07 | **26** |

over 141 instances of RandGraph with $k$=5; see Table 9. We also apply QuAnt to these problem instances. We solve the resulting QUBO with QA and find the average probability to be $26\%$ with a standard deviation of $18\%$, better than any method from the QGM paper (Seelbach Benkner et al., 2020).

This difference is not surprising since we construct our method such that we only use trivial embeddings and do not need to apply the `minorminer` heuristic (Cai et al., 2014). Because of that, for RandGraph with $k = 5$, our method needs only 15 physical qubits while their baseline and row-wise methods need 89 qubits, on average, and a chain length of 4; their *Inserted* method needs, on average, 39 qubits and a chain length of 3 on D-Wave Advantage. Note that a heuristic search for better penalty parameters, as in Q-Sync (Birdal et al., 2021), could give rise to better results for the methods from QGM (Seelbach Benkner et al., 2020) in Table 9. However, the corresponding embeddings would still be problematic. Directly using the binary encoding for permutations (Gaitan & Clark, 2014) requires additional qubits because the problem would a priori not be quadratic.

## E SOLUTION QUALITY OF SA AND QA

In the main paper, we show that training with QA yields better performance than training with SA. Here, we analyse the quality of the solutions found by both techniques further. Fig. 7 contains histograms that depict the output of the quantum annealer and the two different simulated annealing solvers from `neal` (D-Wave Systems, Inc., 2022b) and from `dimod` (D-Wave Systems, Inc., 2022a). In contrast to the solver from `dimod`, `neal` is highly optimised for performance, so we used it for our experiments.

We focus our analysis on the number of sweeps in SA, *i.e.,* the number of steps in the 'cooling' schedule. We observe that it strongly influences the quality of the second-best solution.

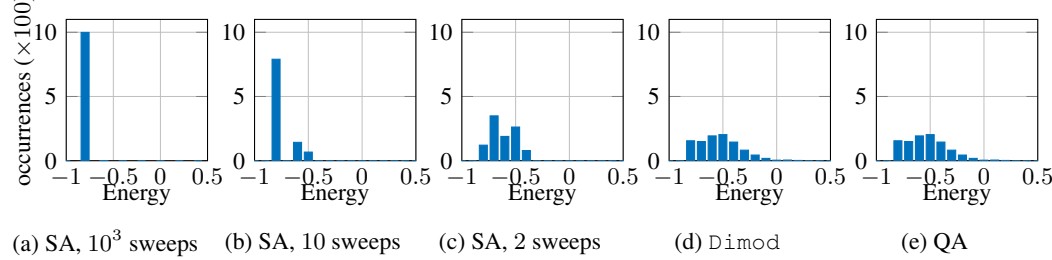

(a) SA, $10^3$ sweeps  (b) SA, 10 sweeps  (c) SA, 2 sweeps  (d) `Dimod`  (e) QA

Figure 7: Energy histograms of (ideally optimal) $10^3$ samples for SA (different number of sweeps and dimod) and QA on one instance of RandGraph with $k$=5 after 450 epochs training.

Table 10 illustrates this by averaging the fraction of the second-best energies over 141 instances and analysing 1000 samples from different solvers. We see that the quantum annealer produces the second-best samples with the lowest energies.

Note, however, that we do not claim that this is an intrinsic general advantage of QA over SA, but merely that in our setting, QA outperforms SA. Still, prior work (Willsch et al., 2020) also reaches the conclusion that quantum annealing has much potential for finding reasonable near-optimal solutions.

The `dimod` sampler also produces second-best solutions with low energies but is computationally expensive (D-Wave Systems, Inc., 2022a). This is, perhaps, because many non-optimal solutions are produced compared to the implementation from `neal`.

Table 10: Second-best energies of SA relative (in %) to the second-best energies of QA. We report the mean and std. deviation over 141 instances. The higher the better.

| SA (`neal`), $10^3$ sweeps | SA (`neal`), 10 sweeps | SA (`dimod`) |
|---|---|---|
| 94.2$\pm$ 10.3 | 86.0 $\pm$ 19 | 99.8 $\pm$ 0.9 |

## F  3D ROTATION ESTIMATION

### F.1  THREE STAGES (EULER ANGLES)

We obtain improved results when regressing three Euler angles one after another compared to direct regression of three angles. Thus, we use one stage per angle, *i.e.,* with one network per stage. The training setup is as follows. We feed the first network with problem instances where $\alpha, \beta, \gamma \neq 0$. The network then regresses $\alpha$. We feed the second stage network with problem instances, where $\alpha = 0$ and $\beta, \gamma \neq 0$. Subsequently, the network regresses $\beta$. Lastly, we feed the third network with problem instances, where $\alpha, \beta = 0$ and $\gamma \neq 0$. Here it regresses $\gamma$. During test time, the first network determines $\alpha$, which is then applied to the input. The updated input is re-encoded before the second stage, which outputs $\beta$. Finally, with $\alpha$ and $\beta$ applied already, the third stage regresses $\gamma$. Fig. 8 shows how the three different networks regress the angles and how the solution progresses towards the final one.

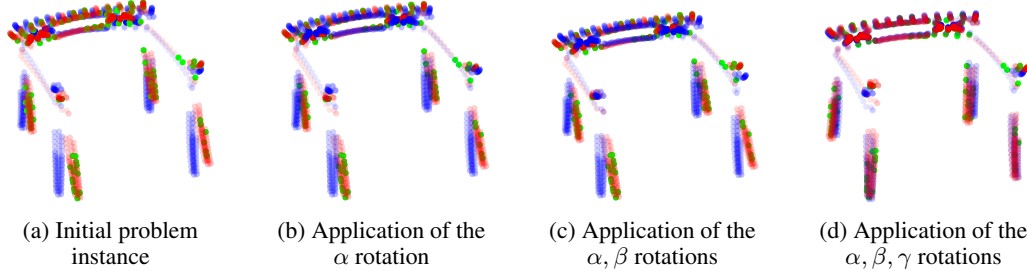

| (a) Initial problem instance | (b) Application of the $\alpha$ rotation | (c) Application of the $\alpha, \beta$ rotations | (d) Application of the $\alpha, \beta, \gamma$ rotations |
|---|---|---|---|

Figure 8: Visualisation of the different steps of our rotation-estimation network. We show (blue) the original 3D point cloud and (red) the rotated point cloud. The green points are points of the red point cloud with unknown correspondences. Here, 10% of the correspondences are unknown.

### F.2  VARIANCE ACROSS RUNS

To better judge the stability under different random seeds, we repeat the main experiment from the paper three times for our QuAnt method and each baseline. In Table 11, we report the mean and std. deviation of the median. Here, similar to the results from the main paper, we outperform the Diag and Pure baselines in all but one setting.

Table 11: Comparison to general baselines on rotation estimation. We report the mean of the per-experiment *median* and std. deviation across experiments for three different random seeds.

|  | **Ours** | Diag | Pure |
|---|---|---|---|
| $L=3, H=32$ | 6.0 $\pm$ 3.6 | **5.3 $\pm$ 1.1** | 7.0 $\pm$ 1.0 |
| $L=3, H=78$ | **4.0 $\pm$ 1.0** | 5.0 $\pm$ 0.0 | 7.0 $\pm$ 0.0 |
| $L=5, H=32$ | **3.7 $\pm$ 1.2** | 5.0 $\pm$ 0.0 | 16.3 $\pm$ 7.2 |
| $L=5, H=78$ | **3.7 $\pm$ 0.6** | 5.0 $\pm$ 0.0 | 9.0 $\pm$ 1.0 |

### F.3  TRAINING ON NOISY DATA

Table 12 shows how our method performs on noise-free and noisy test data after training on noisy data. We observe that the noisy training data appears to negatively affect the training and its performance drops, while Diag improves and Pure remains unchanged.

### F.4  QUALITATIVE COMPARISON ON NOISY TEST DATA

Fig. 9 visualises differences between our solution after training on noise-free data and Procrustes alignment on a problem with noisy data (unknown correspondences).

Table 12: Robustness to varying amounts of incorrect test-time correspondences in rotation estimation. We report the mean/median error for $L=3$, $H=32$. The first column specifies the percentage of incorrect correspondences at test time.

(a) Training without noise

| % | **Ours** | Procrustes | Diag | Pure |
|---|---|---|---|---|
| 0 | 3.9 / 4.0 | **0.0 / 0.0** | 5.6 / 6.0 | 8.1 / 8.0 |
| 1 | **3.4 / 3.0** | 5.8 / 3.0 | 5.7 / 6.0 | 8.2 / 8.0 |
| 5 | **3.4 / 3.0** | 25.7 / 13.0 | 6.0 / 6.0 | 8.2 / 8.0 |
| 10 | **3.2 / 3.0** | 43.8 / 21.0 | 6.2 / 6.0 | 8.2 / 8.0 |
| 15 | **3.5 / 3.0** | 64.7 / 58.0 | 6.2 / 6.0 | 8.2 / 8.0 |
| 20 | **3.7 / 3.0** | 75.3 / 79.0 | 5.8 / 6.0 | 8.2 / 8.0 |

(b) Training with 10% incorrect correspondences

| % | **Ours** | Procrustes | Diag | Pure |
|---|---|---|---|---|
| 0 | 4.4 / 4.0 | **0.0 / 0.0** | 4.6 / 5.0 | 8.1 / 8.0 |
| 1 | **4.3 / 4.0** | 5.8 / 3.0 | 5.1 / 5.0 | 8.3 / 8.0 |
| 5 | **5.0 / 5.0** | 25.7 / 13.0 | 5.1 / 5.0 | 8.2 / 8.0 |
| 10 | **5.2 / 5.0** | 43.8 / 21.0 | 5.2 / 5.0 | 8.2 / 8.0 |
| 15 | 5.2 / 5.0 | 64.7 / 58.0 | **5.0 / 5.0** | 8.2 / 8.0 |
| 20 | 5.6 / 6.0 | 75.3 / 79.0 | **4.9 / 5.0** | 8.2 / 8.0 |

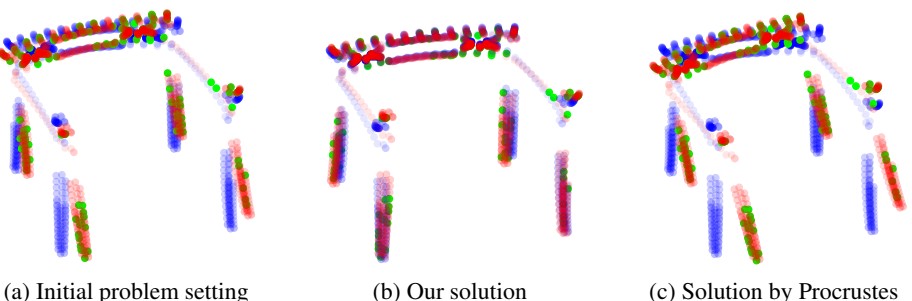

(a) Initial problem setting     (b) Our solution     (c) Solution by Procrustes

Figure 9: Comparison of initial input rotation, our method and Procrustes. The initial 3D point cloud is blue and the rotated one is red, where the unknown correspondences are displayed in green. Here, 10% of the correspondences are unknown.

## F.5 COMPARISON TO AQM (GOLYANIK & THEOBALT, 2020)

QuAnt can estimate 3D rotations with known point matches. However, AQM (Golyanik & Theobalt, 2020) would require 81 densely connected logical qubits, which is not supported by the current quantum-hardware generations. Hence, we cannot compare against AQM for this problem setting and instead only compare to classical methods such as Procrustes.

## G 2D POINT SET REGISTRATION

### G.1 SETTING DETAILS

We encode the point set registration instances similar to Golyanik *et al.* (Golyanik & Theobalt, 2020). As the correspondences between the template and the reference in point set registration are not known, we use $k$-nearest neighbours to find possible correspondences. Our network is trained with three nearest neighbours per each template point.

### G.2 VARIANCE

In Table 13, we report the mean median. Here, we outperform the baselines in all cases. In nearly all setups, we outperform the baselines, and only for one case, we are on par with the Pure baseline. Still, even in that case, QuAnt performs more consistently, as evidenced by its lower std. deviation. These experiments show that we consistently outperform the baselines and the performance is not dependent on the random seed.

Table 13: Comparison of QuAnt to general baselines on point set registration. We report the mean of the per-experiment *median* and std. deviation across experiments for three different random seeds.

| | **Ours** | Diag | Pure |
|---|---|---|---|
| $L=3, H=32$ | **4.8 ± 0.6** | 6.8 ± 0.2 | 5.8 ± 0.9 |
| $L=3, H=78$ | **3.7 ± 0.3** | 5.1 ± 0.4 | 4.8 ± 0.3 |
| $L=5, H=32$ | **4.6 ± 0.1** | 7.1 ± 1.0 | 7.1 ± 1.6 |
| $L=5, H=78$ | **3.4 ± 0.1** | 4.9 ± 0.0 | 7.9 ± 2.7 |

Table 14: Robustness to varying amounts of uniform noise. We report the mean/median error for $L{=}3$, $H{=}32$. The first column specifies the range of the added uniform noise, in %, of the maximum extent of the point cloud.

| % | **Ours** | Diag | Pure | AQM |
|---|---|---|---|---|
| 0 | *7.4 / 4.2* | 11.3 / 7.0 | 7.8 / 6.6 | **4.3 / 2.6** |
| 5 | *6.7 / 3.8* | 11.7 / 7.3 | 8.0 / 6.8 | **4.5 / 2.9** |
| 10 | *7.2 / 4.5* | 12.2 / 6.8 | 8.6 / 7.0 | **5.6 / 3.8** |
| 15 | *8.2 / 4.9* | 12.6 / 6.8 | 9.7 / 8.0 | **5.6 / 3.8** |
| 20 | *11.0 / 6.0* | 13.9 / 8.2 | 14.3 / 10.4 | **5.9 / 3.3** |

### G.3 NOISE

In addition to the experiments in the main paper that uses the largest architecture, we also test the noise resistance on the smallest network setup with $L = 3$, $H = 32$; see Table 14. Here, while QuAnt is better than the baselines, we do not outperform AQM. However, *by construction, AQM is an upper bound for our method as the matrix introduced by Golyanik et al. (Golyanik & Theobalt, 2020) is the same as our input into the network, but it gets directly solved by the QA.*

### G.4 QUALITATIVE ABLATION RESULTS

In addition to the quantitative loss ablation in the paper, we visualise the effect of the losses here. In Figure 10, the full loss results in a nearly ground-truth rotation. However, if we leave out $L_\text{gap}$ or $L_\text{MLP}$ during training, a significant reduction in rotational accuracy is visible.

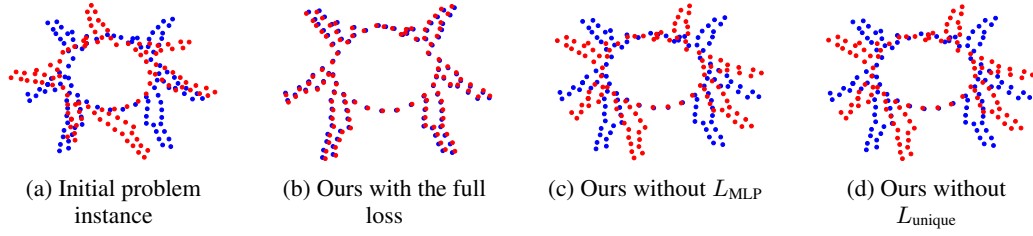

| (a) Initial problem instance | (b) Ours with the full loss | (c) Ours without $L_\text{MLP}$ | (d) Ours without $L_\text{unique}$ |
|---|---|---|---|

Figure 10: Qualitative loss ablations. We show the original point cloud (blue) and rotated point cloud (red). Removing either $L_\text{unique}$ or $L_\text{mlp}$ leads to significantly worse results.

### G.5 FAILURE CASE

We continue our analysis with failure cases. Results in the main paper show that an increasing input angle leads to a reduction in the accuracy of our regressed angle. This can be traced back to our problem-instance encoding as Golyanik *et al.* (Golyanik & Theobalt, 2020) mention that an increasing angle makes finding the correspondences more error-prone. Therefore, an imperfect input encoding makes it is also more likely for us to regress wrong angles.

Similar to most prior work on point set registration, nearly symmetric shapes can be difficult, as most points can be nearly perfectly aligned even with wrong rotations. Fig. 11 contains such a failure case. The initial angle of this problem instance, $50.8°$, is relatively large for our setup, and the shape (which looks like the silhouette of a fish) is nearly rotationally symmetric. In cases like this, our method has difficulties regressing the correct rotation after a single QUBO sampling.

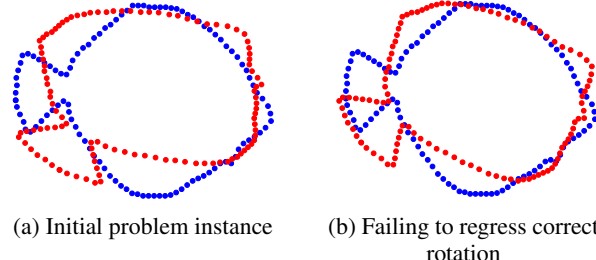

(a) Initial problem instance    (b) Failing to regress correct
                                     rotation

Figure 11: Example of a failure case in point set registration. We show the initial image (blue) as well as the rotated image (red).

