# OpenReview forum: "QuAnt: Quantum Annealing with Learnt Couplings"
_ICLR.cc/2023/Conference — ICLR 2023 notable top 25%_

### Official Review · Reviewer_2Ews · 2022-10-24

**Confidence:** 4
**Correctness:** 3
**Technical Novelty And Significance:** 4
**Empirical Novelty And Significance:** 3
**Recommendation:** 8

**Clarity, Quality, Novelty And Reproducibility:**

- The paper is clear and well-written and the approach seems novel and innovative.

- Reproducibility: the appendices seem to provide the information needed for reproducing the results including the NN and training implementation details and the encoding of the different problems. The supplementary material contains problem instances. Code is not provided with the submission however the authors mention it will be released.

**Strength And Weaknesses:**

Strengths:
* Innovative approach for learning QUBO formulations of problems to be solved on quantum annealers.
* Unlike previous work on quantum computer vision, the proposed approach is general and not problem-specific.
* Experiments on graph matching and point set alignment show the approach is competitive with previous quantum SOTA approaches while requiring fewer qubits.


Weaknesses:
* Learned QUBO can be more sensitive to distribution shift than hand-crafted QUBO design in SOTA approaches. While the noise experiments are interesting it would be valuable to design experiments to compare the proposed approach and the hand-crafted QUBO approaches in terms of generalization to out-of-distribution test instances.
* Learned QUBO formulations can be less interpretable compared to hand-crafted QUBO formulations. In particular, when designing QUBO we often convert problem constraints to penalty terms in the objective. Knowing the penalty terms and their weight can provide some guarantees on the obtained solutions with respect to the satisfaction of domain-specific constraints. While learning QUBO may involve implicitly learning such constraints from data, we have no way of knowing what constraints have been learnt (in human-understandable mathematical notation as often done in the hand-crafted QUBO formulations).
* Experiments do not show significant gains in performance over SOTA. However, I think that obtaining competitive results using fewer qubits and in a problem-independent approach is impressive.

Minor question: How did you create mini-batches for training? Did you accumulate the results of multiple runs with the QA hardware into mini-batches?


**Summary Of The Paper:**

The paper presents a new approach for learning QUBO formulation for problems to be solved on quantum annealers using machine learning in comparison to previous work on quantum computer vision that relied on hand-crafted QUBO formulations. The paper proposes learning to regress a coefficient matrix for QUBO using contrastive loss based on the energies of solutions found the quantum annealer (or other QUBO solvers). Experiments on different computer vision problems show competitive results with previous quantum SOTA while requiring fewer qubits.


**Summary Of The Review:**

I think this is an interesting paper that presents a novel and innovative approach with good experimental results. The main weaknesses include the fact that such formulations are learned and may therefore not generalize well or maintain similar guarantees as hand-crafted QUBO formulations, as well as the relatively modest gains in performance compared to SOTA. Despite the weaknesses, I think this is a nice contribution to the research on quantum computer vision (with potential impact beyond computer vision due to the generality of the approach).

---

> ### Author Response · Authors · 2022-11-08
> **Reply to Reviewer 2Ews**
>
> **Learned QUBO can be more sensitive to distribution shift than hand-crafted QUBO design in SOTA approaches. While the noise experiments are interesting it would be valuable to design experiments to compare the proposed approach and the hand-crafted QUBO approaches in terms of generalization to out-of-distribution test instances.**
>
> Since our method is learning-based, it has the expected difficulties when testing on out-of-distribution samples, as mentioned in the Limitations. Out-of-distribution generalisation is a challenging and important question that is interesting to look at in future work. For example, invariance to certain symmetries could be incorporated at the level of the input problem encoding, which would help with generalisation beyond what naive learning would yield.
>
> **Learned QUBO formulations can be less interpretable compared to hand-crafted QUBO formulations. In particular, when designing QUBO we often convert problem constraints to penalty terms in the objective. Knowing the penalty terms and their weight can provide some guarantees on the obtained solutions with respect to the satisfaction of domain-specific constraints. While learning QUBO may involve implicitly learning such constraints from data, we have no way of knowing what constraints have been learnt (in human-understandable mathematical notation as often done in the hand-crafted QUBO formulations).**
>
> This is a very promising idea and we hint at this kind of follow-up work in our discussion of future work: "Although our focus is on a general design in this paper, our core idea of learning QUBOs can be specialised to any given problem type by designing a more specific network architecture and losses that capture priors for the problem type". If certain soft constraints are desired, they can be added on top of a learned "data" QUBO and might not even need to be learned at all, as they might be independent of the specific problem instance.
>
> **Experiments do not show significant gains in performance over SOTA. However, I think that obtaining competitive results using fewer qubits and in a problem-independent approach is impressive.**
>
> We appreciate that the reviewer agrees with the trade-off we explore in this work: Our proposed approach is competitive with the quantum state of the art, is problem-independent, and needs fewer qubits.
>
> **Minor question: How did you create mini-batches for training? Did you accumulate the results of multiple runs with the QA hardware into mini-batches?**
>
> The second half of Sec. 3.3 describes this. We can put multiple smaller QUBOs as different blocks into the larger, overall QUBO by simply not having any couplings between these blocks. Since this allows us to anneal multiple problems in parallel without compromising on the solution quality of each problem instance, we do not need multiple runs.

---

> > ### Comment · Reviewer_2Ews · 2022-11-28
> > **Thank you for your response**
> >
> > Thank you for your response

---

### Official Review · Reviewer_12QQ · 2022-10-25

**Confidence:** 4
**Correctness:** 4
**Technical Novelty And Significance:** 3
**Empirical Novelty And Significance:** 3
**Recommendation:** 6

**Clarity, Quality, Novelty And Reproducibility:**

The quality and clarity of paper writing is good.
The idea is original and shows advantages.

**Details Of Ethics Concerns:**

No ethics concerns

**Strength And Weaknesses:**

Strengths: The idea of learning and updating QUBO forms is novel. The new neural meta-learning approach is applied to the QUBO on modern QA. The paper shows diverse and detailed experimental results. The paper also runs experiments on the D-Wave real quantum
machines to evaluate their method.

Weakness:
It seems the method requires the classical part to finish the whole back-propagation pass. How will that scale with the number of qubits? will that be the bottleneck of the training efficiency?

**Summary Of The Paper:**

The paper studies the problem of classical-quantum hybrid learning to solve computer vision (QCV) problems. The model contains classical neural network layers and a quantum annealer. The whole system is trained with gradient back-propagation. The experiments on graph matching and point set registration show effectiveness of the proposed method with higher accuracy than baselines.

**Summary Of The Review:**

Interesting meta-learning method for QUBO problems for quantum computer vision, with experiments results on real D-Wave quantum machines.

Thanks for the response! I have read it and would like keep the score.

---

> ### Author Response · Authors · 2022-11-08
> **Reply to Reviewer 12QQ**
>
> **It seems the method requires the classical part to finish the whole back-propagation pass. How will that scale with the number of qubits? will that be the bottleneck of the training efficiency?**
>
> Yes, the classical part uses a standard forward and backward pass. Their runtime scales quadratically in the number of qubits since we need to regress the entries of the $n\times n$ QUBO matrix, where $n$ is the number of qubits. Since the QUBO matrix needs to be densely constructed somehow in any case to use the quantum annealer, it is inherently impossible to improve over this time complexity.
>
> The lack of truly direct and easy access to quantum annealer hardware is the main bottleneck of the training efficiency.  On quantum annealers, the overheads associated with cloud API calls to the D-Wave quantum computer and its imperfect availability slow down training in practice. On classical hardware, we use exhaustive search or SA, which are less efficient than QA at solving QUBOs and hence bottleneck the training.

---

### Official Review · Reviewer_kb2o · 2022-10-29

**Confidence:** 3
**Correctness:** 3
**Technical Novelty And Significance:** 2
**Empirical Novelty And Significance:** 3
**Recommendation:** 6

**Clarity, Quality, Novelty And Reproducibility:**

About Organization & Presentation:
- The presentation is succinct and cogent. Abundant visual demonstrations are provided to aid in comprehension. The results are presented clearly in tables and graphs.
- The paper is organized in a coherent and logical manner.

About Novelty:
- A new neural meta-learning approach to obtain QUBO forms executable on modern quantum annealing algorithms for computer-vision problems.
-This paper is not very novel in its technicality, nor in its application of MLP. Nonetheless, it is commendable to take a general problem by learning its form and transforming it into the ones we have solutions to (rather than attempting to solve it directly).

**Strength And Weaknesses:**

Pros:
- The idea of finding the forms of QUBO with gradient is interesting.
- A wide variety of problems can be tackled successfully and competitively by the proposed general quantum approach.
- The proposed algorithm finds formulation with fewer qubits than state-of-the-art.

Cons:
- The selected baselines are not strong enough. It is not clear how the selected baselines, Diag and Pure measure the quality of the solution. The author should compare to state-of-the-art.
- The comparisons with specialized methods are limited.
- The synthetic data are relatively small, e.g., random matrix 4 by 4 and 5 by 5.

**Summary Of The Paper:**

This paper tackles the problem of generating QUBO forms from an unspecified problem instance. It learns a low-energy formulation of quadratic unconstrained binary optimization problems with multilayer perceptron and gradient backpropagation, instead of problem-specific analytical derivations.  After learning a QUBO form, they solve it with existing quantum annealing techniques. Their model excels previous models on some benchmarks both in computational expenditure and performance.

Challenge:

Depth: A new training strategy for neural methods with backpropagation, independent of the solver.
Results:
-Task: graph matching, 2D point cloud alignment, and 3D rotation estimation.
-Metric: Accuracy, and the number of qubits of the forms obtained by the proposed algorithms.

**Summary Of The Review:**

About Challenge:
- Obtaining suitable QUBO forms in computer vision remains challenging and currently requires problem-specific analytical derivations.
- The MLP is trained with the results from AQM, resulting in its performance bounded by AQM. Given such deficiency, is it really fair to conclude that QUBO learning is capable to deal with a general problem?
- The results are not impressive. Their MLP surprisingly fails to beat binary regression (“pure”) in some cases. Is it an isolated incident that QUBO learning beats some SOTA?

About Contribution:
-The paper contributes to the area by looking beyond the traditional approach of deriving and solving QUBO from a specific problem. It addresses a bigger picture concerning how to identify and classify a mission. By aiming at generality, it extends the application of quantum annealing as well.

---

> ### Author Response · Authors · 2022-11-08
> **Reply to Reviewer kb2o**
>
> **The selected baselines are not strong enough. It is not clear how the selected baselines, Diag and Pure measure the quality of the solution.**
>
> For all methods, we use the same metrics to measure the quality of the solution.
>
> We also refer to our reply to the next question.
>
> **The author should compare to state-of-the-art. The comparisons with specialized methods are limited.**
>
> We do compare to the relevant quantum state of the art, which is the relevant reference point since quantum methods are still very much in their infancy and even specialised quantum methods are usually not competitive with classical methods.
>
> We also emphasise that the specialised methods are only included for reference. While desirable, we believe that our general method cannot be expected to outperform specialised methods, quantum or not. Our competitive performance against the quantum state of the art is thus very encouraging.
>
> **The synthetic data are relatively small, e.g., random matrix 4 by 4 and 5 by 5.**
>
> The problem sizes for all three problem classes we consider are only small in comparison to _classical_ methods. We emphasise that the problem sizes are _on par_ with what state-of-the-art specialised quantum methods can handle, which is the relevant reference point.
>
> **This paper is not very novel in its technicality, nor in its application of MLP. Nonetheless, it is commendable to take a general problem by learning its form and transforming it into the ones we have solutions to (rather than attempting to solve it directly).**
>
> MLPs are of course widely used in computer vision and contrastive losses have been used before in other contexts to circumvent the need for differentiability. We agree that our core contribution is not on that technical level but on the conceptual level: deep learning has not previously been applied to QUBO regression for an entire problem class, especially not in a problem-independent manner. This is particularly relevant as QUBOs are the main interface for using adiabatic quantum computing. We believe that our work opens up numerous avenues for future work that have hitherto not even been considered. We also believe it speaks to the conceptual novelty of our method that it can make do with simple and elegant technical components, as no prior work has even tackled our problem setting.
>
> **The MLP is trained with the results from AQM, resulting in its performance bounded by AQM. Given such deficiency, is it really fair to conclude that QUBO learning is capable to deal with a general problem?**
>
> We require a problem encoding for a general problem class. AQM hand-crafts a QUBO for point set registration. We merely take this hand-crafted QUBO as the input _because it provides us with a problem encoding for this problem class_. We emphasise that we do _not_ take the results from AQM as input.
>
> We also refer to our reply to the next question.
>
>
> **The results are not impressive. Their MLP surprisingly fails to beat binary regression (“pure”) in some cases. Is it an isolated incident that QUBO learning beats some SOTA?**
>
> Most of the time, our method does outperform the Pure baseline.
>
> We also note that we did not pick the problem classes based on whether they work. We rather report results on all problem classes we experimented with. They can thus be considered somewhat random examples of problem classes. It is therefore not unexpected that we do not achieve great performance in all cases. This just offers a better impression of the performance that can be expected on other problem classes than if we only reported on cherry-picked problem classes. It is thus not an isolated, cherry-picked incident that our method beats the state of the art.

---

### Author Response · Authors · 2022-11-08
**General Comment**

We are thankful for the encouraging and helpful feedback and are happy to see that our work is well received. We appreciate that the reviewers find the paper interesting, well written and the method well evaluated; that it could have a potential impact beyond computer vision. We next address the raised concerns in individual replies.

---

### Decision · Program_Chairs · 2023-01-20

**Decision:**

Accept: notable-top-25%

**Justification For Why Not Higher Score:**

The synthetic data are relatively small, and the experiments do not show significant gains in performance over SOTA.

**Justification For Why Not Lower Score:**

All reviewers acknowledged that the proposed approach is novel with clear advantages.

**Metareview: Summary, Strengths And Weaknesses:**

Summary: The paper studies the problem of classical-quantum hybrid learning to solve computer vision (QCV) problems. The model contains classical neural network layers and a quantum annealer. Experiments on different computer vision problems show competitive results with previous quantum SOTA while requiring fewer qubits.

Strengths: The idea of learning and updating QUBO forms is novel. The new neural meta-learning approach is applied to the QUBO on modern QA. The proposed algorithm finds formulation with fewer qubits than state-of-the-art.

Weakness: Learned QUBO can be more sensitive to distribution shift than hand-crafted QUBO design in SOTA approaches.

**Note From Pc:**

if the above contains the word "oral" or "spotlight" please see: "oral" presentation means -> notable-top-5% and "spotlight" means -> notable-top-25%. As stated in our emails, we are disassociating presentation type from AC recommendations